# Recent Developments in the Biological Activities, Bioproduction, and Applications of *Pseudomonas* spp. Phenazines

**DOI:** 10.3390/molecules28031368

**Published:** 2023-02-01

**Authors:** Bruno Serafim, Ana R. Bernardino, Filomena Freitas, Cristiana A. V. Torres

**Affiliations:** 1Laboratory i4HB—Institute for Health and Bioeconomy, School of Science and Technology, NOVA University Lisbon, 2825-149 Caparica, Portugal; 2UCIBIO—Applied Molecular Biosciences Unit, Department of Chemistry, School of Science and Technology, NOVA University Lisbon, 2825-149 Caparica, Portugal

**Keywords:** phenazines, *Pseudomonas* spp., bioprocess

## Abstract

Phenazines are a large group of heterocyclic nitrogen-containing compounds with demonstrated insecticidal, antimicrobial, antiparasitic, and anticancer activities. These natural compounds are synthesized by several microorganisms originating from diverse habitats, including marine and terrestrial sources. The most well-studied producers belong to the *Pseudomonas* genus, which has been extensively investigated over the years for its ability to synthesize phenazines. This review is focused on the research performed on pseudomonads’ phenazines in recent years. Their biosynthetic pathways, mechanism of regulation, production processes, bioactivities, and applications are revised in this manuscript.

## 1. Introduction

Phenazines are a large group of heterocyclic, colored, and nitrogen-containing compounds of biological and chemical origin with great stability in natural environments. These natural compounds were first referred to in 1859 by Fordos [1,2], who observed a blue–green pigment responsible for the coloration of the “blue pus” associated with severe wounds resulting from surgical procedures. This pigment was named pyocyanin (**1**, Table 1) and 65 years later it was identified as a phenazine derivative [3,4]. Since then, over 150 natural phenazine derivatives and more than 6000 synthetic derivatives with wide-ranging bioactivities have been reported [5,6].

The position and type of functional groups in the phenazines’ molecules dictate their chemical, physical, and biological properties that encompass antibiotic, antifungal, insecticidal, antitumor, antimalarial, and antiparasitic activities. This wide bioactivity spectrum is conferred by the high redox activity of phenazine molecules and their ability to reduce molecular oxygen to reactive oxygen species (ROS) [2,7].

Phenazine-producing bacteria are found all over nature in association with plant and animal hosts that live in terrestrial, freshwater, and marine habitats, presenting an important role in bacterial physiological processes, namely, in biofilm formation and in iron reduction. These metabolites play an important role in the ecosystems in which their producers and hosts live, such as in the survival and persistence of rhizobacteria [7,8].

The *Pseudomonas* genus, which is well-known for its ability to produce multiple secondary metabolites, includes producers of almost one-third of all known phenazines. Among them, 1-hydroxyphenazine (**2**, Table 1), 2-hydroxyphenazine (**3**, Table 1), phenazine-1-carboxylic acid (**4**, Table 1), and phenazine-1-carboxamide (**5**, Table 1) are the most common derivatives [9]. Phenazine-1,6-dicarboxylic acid (PDC) and PCA act as ‘‘core’’ phenazines that strain-specific genes convert to a different type of phenazine. These phenazines can suffer other natural modifications and be converted into different derivatives [7,9]. Hence, depending on the strain and the environment, PDC and PCA can be converted into several different phenazines (Table 1).
molecules-28-01368-t001_Table 1Table 1Different phenazine derivatives associated with their producing bacterial strain.CompoundPhenazine DerivativeBacterial StrainReference**1**Pyocyanin (PYO)*Pseudomonas aeruginosa*[10]**2**1-hydroxyphenazine (1-OH-PHZ)*Pseudomonas aeruginosa*[10]**3**2-hydroxyphenazine (2-OH-PHZ)*Pseudomonas chlororaphis*[11]**4**phenazine-1-carboxylic acid (PCA)*Pseudomonas chlororaphis*[11]**5**phenazine-1-carboxamide (PCN)*Pseudomonas aeruginosa*;*Pseudomonas chlororaphis*[11,12]


Natural bacterial phenazines are highly studied because of their importance in interactions with other microorganisms, and also due to their beneficial interactions with plants and animals. Thus, besides the increase in production yields, its isolation from cultivation broths and characterization are being studied. This aims to achieve efficient strategies to recover the product, as well as to elucidate the structure and function to find suitable applications. This review will bring together numerous key aspects of phenazines, focusing on those produced by *Pseudomonas* spp., ranging from the biological origin of the producing strains and the properties of these versatile secondary metabolites to their uses as biocontrol agents in agriculture and human health.

## 2. Phenazine-Producing Strains

Phenazines are produced by diverse bacteria mainly from *Pseudomonas* spp. and *Streptomyces* spp., but other bacteria have also been reported, isolated from terrestrial and marine environments, namely, *Actinomycetes* spp., *Vibrio* spp., *Burkholderia* spp., and *Brevibacterium* spp., among others [5,13,14].

Among all phenazine producers, *Pseudomonas* is the most studied genus at the metabolic and genomic level for that secondary metabolite production, especially fluorescent *Pseudomonas* [5,11,12]. A diverse phenazine production is observed for each strain, depending on the biosynthetic capabilities for phenazine derivatization, but those isolated from *Pseudomonas* spp. are mostly simple hydroxyl and carboxyl structures [15]. *P. chlororaphis* has been reported to produce phenazine-1-carboxylic acid (PCA), 2-hydroxyphenazine-1-carboxylic acid (2-OH-PCA) (Figure 1), 2-hydroxyphenazine (2-OH-PHZ) (Figure 2), and phenazine-1-carboxamide (PCN) [16,17]. For instance, *P. choloraphis* HT66 and PCL1391 produces PCN, while *P. chlororaphis* GP72 produces PCA, 2-OH-PHZ like *P. chlororaphis* 30-84 which is also reported for 2-OH-PCA production [11,18,19,20,21,22]. *P. aeruginosa* can produce pyocyanin (PYO), PCA, PCN, 1-hydroxyphenazine (1-OH-PHZ), and Aeruginosin A and B [5,23]. Examples of these are *P. aeruginosa* PA1201 and M18 which produces PCA, like strain LV which, additionally, produces PCN, and *P. aeruginosa* JY21 which produces PYO [23,24,25,26,27,28]. *P. fluorescens*, strains fp-5 and 2-79, produce only PCA [11,29,30].

Phenazine production by wild-type strains present low yields to be used in large-scale applications, namely, as a biopesticide in agriculture. To improve phenazine production, based on genetic knowledge, recombinant engineered microorganisms are being developed and investigated as potential cell factories, as well as for growth medium optimization [31]. In fact, Peng et al. [12] achieved a PCN production above 9 g/L by engineering a *P. chlororaphis* HT66 strain. Zhou et al. [31] reported a PCA production of 6.4 g/L with an engineered *Pseudomonas* sp. M18 and Jin et al. (2015) have achieved 9.8 g/L of PCA with a *P. aeruginosa* PA1201 engineered strain (named PA-IV), while the production with the wild-type bacteria was reported to be only 180 mg/L (Table 2).

Regarding media optimization, *P. chlororaphis* GP72 was engineered to utilize the glycerol pathway. The two genes, glpF and glpK, from the glycerol metabolism pathway were overexpressed in GP72ANO separately and after were co-expressed, resulting in a production improvement from 729.4 mg/L to 993.4 mg/L. To enhance the glycerol use, the shunt pathway was blocked, resulting in 1493.3 mg/L PCA production [32].

Li et al. [33] tested different strategies to enhance PCN production, namely, the knocking out of negative regulatory factors, the enhancement of the shikimate pathway by gene overexpression and gene knocking, and by using fed-batch cultivation. Finally, an engineered strain of *P. chlororaphis* achieved 11.45 g/L PCN.
molecules-28-01368-t002_Table 2Table 2Phenazine production by *Pseudomonas* sp., including phenazine type and the amount produced.Strain NameStrain CharacteristicsPhenazine TypePhenazine Production (mg/L)Reference*Pseudomonas aeruginosa* JY21Wild typePYO311.1[28]*Pseudomonas aeruginosa* LVWild typePCA112.89[27]PCN177.31*Pseudomonas aeruginosa* PA1201Wild typePCA180[24]*Pseudomonas aeruginosa* PA-IVDelete *phz*S, *phz*M, *phz*H, *pab*B/C, *trp*E, and *pch*A; overexpression of *aro*G and *phz*C1; engineering the promoters of PCA biosynthetic gene clusters and the efflux pump, etc.PCA9882[24]*Pseudomonas* spp. MCC 3145Wild typePYO313.94[34]*Pseudomonas* sp. M18Wild typePCA48.0[35]*Pseudomonas* sp. M18 G*gac*A deficientPCA2597[33]*Pseudomonas* sp. M18 GQ*gac*A deficient; overexpression of *phz* gene clusterPCA6365[33]*Pseudomonas chlororaphis* HT66Wild typePCN424.87[12]*Pseudomonas chlororaphis* HT66 P3∆*lon*Point mutations in 138 genes; deletion of *lon* genePCN9174[12]*Pseudomonas chlororaphis* GP72Wild typePCA22.0[20]2-OH-PHZ4.5*Pseudomonas chlororaphis* GP72ANInactivation of *rpe*A genePCA432/NR[20,36]2-OH-PHZ170/258*Pseudomonas fluorescens* 2-79Wild typePCA1010[37]


## 3. Phenazines Biosynthesis

### 3.1. Metabolic Pathways

Phenazine biosynthesis is derived from the shikimate pathway, which is also known for the production of the amino acids: phenylalanine, tyrosine, and tryptophan in microorganisms [38,39,40]. From this pathway, the branch point for phenazine production is chorismic acid, giving rise to the basic phenazine structure.

The shikimate pathway (Figure 3) starts with the condensation of erythrose-4-phosphate (E4P) and phosphoenolpyruvate (PEP) to form D-arabino-heptulosonate-1-phosphate (DAHP), catalyzed by the enzyme PhzC, phospho-2-dehydro-3-deoxyheptonate aldolase [38,40]. DAHP is then converted into chorismate, the precursor of phenazines. The following step is the conversion of chorismate into 2-amino-4-deoxyhorismic acid (ADIC) by PhzE (anthranilate/para-aminobenzoate synthase). Subsequently, ADIC is hydrolyzed to pyruvate and trans-2,3-dihydro-3-hydroxyanthranlic acid (DHHA) by PhzD, an isochorismate hydrolase. DHHA is then converted to 6-amino-5-oxo-cyclohex-2-ene-1-carboxylic acid (AOCHC) by PhzF [40,41]. The next step is the formation of the phenazine tricycle, catalyzed by enzymes PhzB/A leading to hexahydrophenazine-1,6-dicarboxylic acid (HHPDC) [42,43,44]. This molecule suffers oxidative decarboxylation, giving rise to tetrahydrophenazine-1,6-carboxylic acid (THPCA). THPCA is transformed into 5,10-dihydrophenazine-1-carboxylix acid (DHPCA), catalyzed by PhzG (pyridoxamine 5′-phosphate oxidase) [42,43,44,45]. DHPCA is finally converted into PCA, by spontaneous oxidation by air. Figure 3 illustrates the biosynthetic pathway of *Pseudomonas* genus phenazines.

The production of strain-specific phenazines is carried out by enzymes encoded by genes like *phzH*, *phzM*, *phzO*, and *phzS*, which usually flank the core gene cluster [24,46]. The production of these phenazines is coupled with the biosynthesis of the core molecule PCA. However, the modifications are thought to occur in DHPCA, instead of PCA, due to its higher reactivity [45]. The enzyme PhzM is a phenazine-specific methyltransferase and is reported to catalyze the formation of 5-methylphenazine-1-carboxylate (5MPCA) that is then converted to PYO by PhzS, a flavin-containing monooxygenase [23,46]. These two enzymes are required to obtain PYO and the genes that encode them are reported to be present in *P. aeruginosa* PAO1, for instance [23]. However, when only PhzS is present the phenazine derivative produced is 1-OH-PHZ [23]. PhzH, an asparagine synthase, is the enzyme involved in the conversion of DHPCA into PCN, resulting in a large accumulation of PCN compared with the amount of PCA detected [17,23,47,48]. This enzyme is found in *P. chlororaphis* PCL1391, *P. chlororaphis* HT66, and *P. aeruginosa* PAO1 [49,50]. The production of 2-OH-PCA is mediated by PhzO, a phenazine hydroxylase and afterward, a part of it suffers a spontaneous decarboxylation to form 2-OH-PHZ [20,46,47]. In this process, PhzO is believed to suffer substrate inhibition leading to a low yield of 2-OH-PCA and 2-OH-PHZ [51]. Such a modification pathway is reported for *P. chlororaphis* 30-84 and PCL1391 [46].

Phenazine-producing *Pseudomonas* spp. are reported to produce mainly PCA and PCA derivatives, while phenazine-1,6-dicarbooxylic acid (PDC) is typically produced by *Streptomyces* spp. [34,52]. The exception is *Pseudomonas aeruginosa* strain HRW.1-S3, which was reported to produce PDC when growing in the presence of crude oil as the sole carbon source [52].

In *Pseudomonas* spp. the gene cluster *phzABCDEFG* is responsible for converting the chorismate into PCA and it is highly conserved in all phenazine-producing pseudomonads [5,34,53]. Recent studies suggest that the PCA production by *Pseudomonas* is related to the presence of the *phzA* gene (that presents high homology with the *phzB* gene), whose presence is restricted to this type of phenazine producer, and the expression level of *phzG* gene [44,45]. Guo et al. [44] reported a significant PDC production by a *P. chlororaphis* HT66 *phzA*-disrupted mutant alongside PCA production. Later, Guo et al. [45] reported a spontaneous PDC synthesis by a *P. chlororaphis* GP72AN without the *phzG* gene.

### 3.2. Biological Regulation

In *Pseudomonas* spp., phenazine gene expression and consequent phenazine production is under the control of a two-component system, namely, the quorum-sensing (QS) mechanism and a small RNA (sRNAs) system [54,55,56,57].

In QS, bacteria release small molecules or peptide signals into the environment, which can then interact with bacteria of the same strain and activate the expression of genes under the control of this mechanism (Figure 4). Molecules with diverse structures have been associated with this behaviour, but acyl homoserine lactones (AHLs) are the most relevant for phenazine expression. Monitoring the number of compounds that are QS regulated, it becomes clear that they arise when a minimum cell concentration is reached. Therefore, those signals enable bacteria to sense their own population size and delay the expression of some genes until a specific cell density is achieved [54,55,58].

The most studied and best characterized QS mechanism regulates phenazine production mediated by N-acyl-L-homoserine lactones (AHL), as found in *P. chlororaphis* strains [12,59]. The genes *phzI* encodes for AHL-syntase, and *phzR* encodes for AHL-receptor, which precede the operon *phz*. The enzyme PhzI is responsible for producing the AHLs that accumulate in the external medium and, when it reaches a threshold level, binds to PhzR activating the transcription of the *phz* operon [50,59].

The sRNA signalling is controlled by GacS sensor kinase which is highly important since it gives bacteria the potential to directly understand the environmental signs and then modulate the activities of regulatory mechanisms or control *phz* expression. This occurs when environmental conditions change and the phosphorylation status of the sensor protein is altered through the binding of a small molecule or other environmental signals that activate conformational changes and affects the activity [7,56,57].

The two-component system most studied is the GacS/GacA where the membrane-bound protein GacS is regulated by GacA, a cytoplasmic regulatory protein, promoting the expression of small non-coding RNAs (e.g., RsmZ). These RNAs bind to translational repressors preventing the repression of the core biosynthetic genes *phz* in bacteria such as *P. chlororaphis* [60,61]. However, the regulation can be different among *Pseudomonas* spp. [61,62]. RpeA/RpeB is another two-component system found in *P. chlororaphis*, in which RpeA negatively regulates the *phzR* expression by negative regulation of *pip* and RpeB has the opposite effect [61]. Pip is an activator of phenazine production in *P. chlororaphis* that is also regulated by the GacS/GacA system [61,63]. *P. chlororaphis* is divided into four subspecies, Morohoshi et al. [63] investigated the relation between phenazine production and quorum sensing in the two subspecies that had not been studied yet, namely *P. chlororaphis* subsp. *chlororaphis* and *piscium*. It was found that the disruption of the *phzI* caused no production of phenazine-1-carboxylic acid (PCA) and phenazine-1-carboxamide (PCN). On the other hand, PCA and PCN production were not affected by the disruption of CsaR-CsaI, a second quorum-sensing system identified in *P. chlororaphis* subsp. *aureofaciens* 30-84. Moreover, Morohoshi et al. [63], showed that the PhzI/PhzR quorum-sensing system plays an important role in the production of phenazine derivatives in both strains.

## 4. Phenazine Production

Phenazine production is growth associated with and affected by environmental factors including temperature, culture medium composition, oxygen availability, and pH value [6,22]. In view of this, the improvement of fermentative conditions is being studied to understand and enhance phenazine production. However, different strains have different nutritional needs depending on their environment [26,30]. So, strategies such as the application of statistical methods for growth medium optimization, use of different feeding strategies, or evaluation at a genetic level, and factors that are associated with phenazine production are being studied to improve their biosynthesis [22,24,64,65]. Rij et al. [22] showed that PCN production by *P. chlororaphis* PCL 1391 is induced by some carbon sources, such as glucose, glycerol, and L-pyroglutamic acid. Similarly, other authors found glucose to be the best carbon source for phenazine production by engineered strains of *Pseudomonas* sp. M18 (M18G) and *P. fluorescens* 2-79, or glycerol for *P. fluorescens* fp-5, *Pseudomonas* spp. MCC3145, and the engineered strain *P. chlororaphis* P3∆*lon* [12,64,66,67]. Ethanol complemented with glucose was found to be the best carbon source for PCA production by the *Pseudomonas* sp. M18G and M18GQ strains [28,31]. Regarding nitrogen sources, ammonium ions and amino acids are reported to have a stimulatory effect on phenazine production, with peptone, tryptone, soybean meal, and corn steep liquor being the most significant [12,22,26,29,31,64]. Yuan et al. [65], using a surface response methodology, found the ideal concentrations of glucose and soytone to increase the PCA production by *Pseudomonas* sp. M18Q, the *qscR*-chromosomal inactivated mutant, from 750 mg/L to 1240 mg/L. Similarly, Peng et al. [12] improved the PCN production of the *P. choloraphis* P3∆*lon* mutant nearly three-fold to 9174 mg/L in an optimized medium, in which glycerol, tryptone, and soy peptone were identified as the most significant factors. For the genetically engineered *Pseudomonas* sp. M18G (a *gac*-inactivated mutant), the best results for PCA production were found with soybean meal complemented with soy peptone as a nitrogen source and ethanol complemented with glucose as a carbon source, having achieved 1.98 g/L, near a three-fold increase from the basal medium [26].

Temperature is also an important factor for phenazine production, and some enzymes involved in the phenazine production process are temperature dependent. That is the case of PhzM, which in *P. aeruginosa* PAO1 is expressed at higher levels at 37 °C, and in *Pseudomonas* sp. M18 at 28 °C, meaning that, depending on the temperature, the ratio between PCA and PYO will be different [23]. Similar results were achieved by Bedoya et al. [27] for the production of PCA and PCN by *P. aeruginosa* LV, where at 28 °C is possible to achieve a higher PCN production, than at 32 °C, favoring PCA production. On the other hand, a considerable decrease in PCN production was observed when the temperature decreased from 21 ºC to 16 °C. Moreover, acidic pH was found to negatively affect phenazine production, having observed a dramatic decrease in PCN synthesis when the pH was changed from 7 to 6 [23]. Similarly, the pH value was found to be a significant factor for PYO production by *P. aeruginosa* JY21 in Abo-Zaid et al. [28], being 8.2 as the best value for their process. Cui et al. [30] optimized the pH to 7.2 in a 1 L bioreactor, leading to an increase in AHLs that resulted in a higher PCN production (8.58 ± 0.25 g/L) by *Pseudomonas chlororaphis* H5ΔfleQΔrelA.

Nutrients such as oxygen, ferric iron, phosphate, sulphate, and magnesium were also reported to be significant in phenazine production [22,27,28,64]. Low concentrations of ferric iron, phosphate, and sulphate, as well as high magnesium and oxygen levels (above 1%), were found to reduce PCN levels in *P. chlororaphis* PCL1391 [22]. Jin et al. [64] reported that the downregulation of proteins involved in phosphate transport could result in an increase in PCN production, as well as the upregulation of proteins involved in iron homeostasis. The presence of ferric iron is also reported to enhance the conversion of PCA or DHPCA to 2-OH-PHZ since the reaction is Fe^3+^ dependent [51].

Operational strategies have been also studied to increase phenazine production. A fed-batch strategy was reported by Jin et al. [24] and Li et al. [33] for PCA production by the engineered *Pseudomonas* sp. M18G and *P. aeruginosa* PA-IV, respectively. Li et al. [33] found that glucose fed in a pulse manner to keep the concentration above the inhibitory level favored higher cellular growth. A different strategy was followed by Huang et al. [20] that added exogenous PCA (400 µg/mL) to the cultivation of the engineered *P. chlororaphis* GP72AN and favored PCA and 2-OH-PHZ production. Yue et al. [36] tried a two-stage cultivation strategy involving the use of the reducing agent DTT (Dithiothreitol), to create reducing conditions in the medium favoring a higher production of PCA. Posteriorly, they fed the system with the electron acceptors K_3_[Fe(CN)_6_] and H_2_O_2_ to generate an oxidative environment allowing the activity of PhzO to convert PCA into 2-OH-PHZ. This strategy was found to improve phenazine production, since phenazines are involved in ROS production that can be harmful to the producer itself, affecting phenazine yields. Aloui et al. [67] reported the improvement of co-production of medium-chain-length polyhydroxyalkanoates (mcl-PHA) and extracellular phenazines through a high cell density cultivation of *Pseudomonas chlororaphis* subsp. *aurantiaca* (DSM 19603) in a membrane bioreactor using crude glycerol as a feedstock.

## 5. Extraction and Purification of Phenazines

The most commonly reported method for extraction of phenazines involves the use of organic solvents applied in a cell-free supernatant that can suffer a previous acidification step. Organic solvents such as ethyl acetate, dichloromethane, chloroform, methylene dichloride, toluene, or diethyl ether are the most reported for this step of the purification process. Then, the organic phase where phenazines are dissolved can be evaporated and the dry solids dissolved in acetonitrile or methanol [19,20,22,23,27,29,32,52,68,69,70]. Some authors, after solvent utilization, include a chromatography step with silica gel to enhance the purification [29,71,72,73,74]. However, the use of organic solvents produces hazardous wastes requiring precautions for safe handling [75].

Rane et al. [76] reported a prior step in PCA extraction, the use of a XAD-4 resin column to remove impurities followed by the extraction of the phenazine compound with chloroform from the acidified solution followed by crystallization.

The separation and purification are mostly reported by thin layer chromatography (TLC) and preparative high-performance liquid chromatography (HPLC) which have good separation resolutions [68,71,75,76,77]. However, other techniques are being studied, such as capillary electrophoresis, free flow electrophoresis, and the use of microporous adsorbent resins as alternatives to reduce the price, the use of organic solvents and the time required for the separation [68,70,78,79,80]. Liu et al. [78] applied capillary zone electrophoresis to separate and quantify PCA and 2-OH-PHZ from the culture broth of *P. chlororaphis* GP72, after an extraction step, offering a rapid and sensitive detection. Free flow electrophoresis for low-concentration PCA purification from *Pseudomonas* sp. M18 cultivation broth was reported by Shao et al. [70]. This method, which requires a previous step for phenazine extraction, offers the advantages of a continuous process, a lower cost due to the consumption of the mobile phase, and high product recovery. Bilal et al. [68] reported the separation and purification of PCA from the cultivation broth of the engineered *P. chlororaphis* GP72AN with microporous adsorbent resins using methanol as a desorbing agent. This strategy is presented as a simple and efficient process that can be used for a scale-up process. Zhou et al. [81] developed electrochemical sensors based on laser-induced graphene for real-time monitoring of *P. aeruginosa* phenazines.

## 6. Phenazine Characterization

In recent years, different techniques to characterize phenazines have been studied taking into consideration the properties of phenazines, namely, their fluorescent properties, redox activity, and light adsorption properties.

TLC, HPLC, and UV-visible spectroscopy have been used by several authors to detect and characterize phenazines [13,20,32,69,72,76,79]. Further, structural characterization and identification of phenazines are performed, after extraction, separation, and purification, using mass spectroscopy (MS, liquid, or gas), nuclear magnetic resonance (NMR, ^1^H and ^13^C) (Figure 5), and Fourier Transform Infra-Red spectroscopy (FTIR) (Figure 6) [69,72,73,74,76,78].

Kern and Newman [82] reported the use of HPLC to separate the different phenazines, namely, PYO, PCA, and 1-hydroxyphenazine (1-OH-PHZ), which were then quantified by UV absorption. PCA and PCN were detected by Peng et al. [12] using HPLC with a C18-WR reversed-phase column at 254 nm. The retention times for PCA and PCN were approximately 9.523 and 17.217 min, respectively. Simionato et al. [83] purified and identified PCA produced by *P. aeruginosa* LV strains by extracting it with dichloromethane. Then, the purity degree of the extracted phenazine was determined using reversed-phase HPLC semi-preparative and the structure was confirmed by NMR and electrospray ionization mass spectroscopy.

PCA and 1-hydroxyphenazine were extracted and purified from a cultivation supernatant of *P. aeruginosa* JAAKPA using TLC and column chromatography. Then, the sample was subjected to gas chromatography–mass spectrometry (GC-MS), FTIR (Fourier Transform Infra-Red spectroscopy), and NMR were then identified [78]. Further, LC-MS was used by Yu et al. [18] to identify different phenazines (e.g., PCA, PCN) produced by *P. chlororaphis* 30-84 (Figure 5). Jasim et al. [69] identified a PCA produced by an endophytic *P. aeruginosa* isolated from *Zingiber officinale*. Firstly, they extracted the compound and then it was purified by TLC being obtained from 7 different fractions, then the presence of compounds was measured by UV scan. Further bioactivity was tested and the fraction that showed activity against *Pythyum myriotylum* was subjected to LC-MS (Light Scattering–Mass Spectroscopy) to identify the compound (Appendix A).

Extracellular metabolites were extracted using ethyl acetate and a UV spectrum was recorded to quantify phenazines. Further, the compound was purified using silica-gel column chromatography. The purified compound was characterized and identified using different techniques, namely, FTIR, NMR, and mass spectroscopy (Appendix A). IR spectra (Appendix A) showed peaks related to the aromatic C–H stretching vibrations (2963, 2924, and 2853 cm^−1^). Moreover, an amide carbonyl group was also confirmed. The ^1^H NMR spectrum (Appendix A) of the *P. aeruginosa* MML2212 purified compound showed two doublets, and two protons were also displayed. Further, three aromatic protons appeared as a multiplet in the range. The ^13^C NMR (Appendix A) spectrum of the purified compound showed the presence of a carbonyl carbon of the amide group. Finally, the sample was subjected to mass spectroscopy (Appendix A); the compound was identified as PCN (phenazine-1-carboxamide) [72].

*P. aurantiaca* produced secondary metabolites with antifungal activity. The metabolites were identified using TLC, UV spectra, and MALDI-TOFF spectra and were revealed to be PCA, 2-hydroxyphenazine (2-OH-PHZ), and N-hexanoyl homoserine lactone (HHL) (Appendix A) [71].

## 7. Properties, Applications, and Commercial Products

Phenazines present antifungal and antibacterial activities, being recognized as broad-spectrum antibiotics. Pierson and Pierson [84] reported the ability of phenazines to eliminate the disease of wheat root caused by the fungus *Gaeumannomyces graminis* var. *graminis*.

*Pseudomonas* phenazine species work also as biocontrol promoters of plant health, such as *Macrophomina phaseolina*, *Fusarium graminerum*, and *Rosellinia necatrix. Macrophomina phaseolina* is one of the most virulent phytopathogens that can infect more than 500 plant species, such as soybean, chickpea, or peanut, causing dry root and stem rot, known as charcoal rot. PCA shows a high antifungal activity against this fungus, presenting a MIC_50_ (Minimum Inhibitory Concentration 50%) value of 35 µg/mL [85,86].

Apart from antifungal activity, phenazines may also act as antibacterial agents [87]. Phenazines possess antibiotic activity against, for instance, *Staphyloccoccus aureus*, *Escherichia coli*, and other bacteria [88,89]. D-Alanylgriseoluteic acid, a potent antimicrobial phenazine compound, is a prime example of the antibacterial activity of these metabolites. This phenazine exhibits a MIC_50_ value of 0.25 µg/mL and a MIC_90_ value of 0.5 µg/mL for penicillin-resistant isolates of *Streptococcus pneumoniae*. Both MIC values of penicillin for these isolates were 4 µg/mL [90]. The antibiotic effect is attributed to the redox properties of phenazines and their capacity to promote the formation of toxic reactive oxygen species (ROS).

Several authors reported PCA activity against different fungi (e.g., *Fusarium oxysporum*, *Penicillium expansum*, *Rhizoctonia solani*), produced by different *Pseudomonas* spp. [74,91]. *Pseudomonas* sp., namely, *P. piscium* and *P.aeruginosa*, also produced phenzine 1HP that has activity against fungi, such as *Fusarium graminearum*, *Colletotrichum gloeosporioides*, *Exserohilum turcicum* [92,93,94]. Park et al. [91] reported activity against *Rhizoctonia solani* by phenazine 2-OH-PHZ produced by *P. aurentiaca* IB5-10. Karmegham et al. [95] also reported the antifungal activity of phenazine derivatives isolated from fluorescent pseudomonads (FPs). PCN was detected in the isolates of FPs, showing a prominent antifungal activity against *R. solani* and other tested fungal pathogens.

Antifungal activity was likewise reported against a major human pathogen fungus, *Trichophyton rubrum*, which could be responsible for causing athlete’s foot, jock itch, ringworm, and fingernail fungus infections [5,74]. Phenazines were tested against *Candida* species, which are responsible for candidiasis infections. Morales et al. [96] showed that 5-methylphenazine-1-carboxylic acid (5MPCA) produced by *Pseudomonas aeruginosa* has an antibiotic effect against *Candida albicans.* A synergistic effect between phenazines isolated from *P. aeruginosa* and azoles were observed against several species of genus *Candida*, and, further, no cytotoxicity against human cell lines was observed [74].

Furthermore, it was demonstrated by different authors that phenazines possess anticancer and neuroprotective activity. The anticancer ability is an outcome ensured by known mechanisms, such as polynucleotide interaction, topoisomerase inhibition, and radical scavenging [5,7,15,97]. Polynucleotide interaction occurs because the aromatic phenazine core has structural similarity to known intercalators-metabolites capable of inhibiting DNA replication and several studies of the interactions between phenazines and DNA/RNA have been made. In 1971, Hollstein et al. [98] reported the study of the interaction between *Pseudomonad* pyocyanin, two other phenazines, and various polynucleotides. Kennedy et al. [99] isolated a 5MPCA produced by *Pseudomonas putida*, which exhibited selective cytotoxicity regarding the cell lines of breast and lung cancer. The isolated phenazine inhibits cell viability, DNA synthesis, induced G cell cycle arrest, and apoptosis in cancer cells.

Topoisomerases (I and II) are enzymes responsible for topological changes in the DNA strand during cell division (translation and transcription). Proliferating cells, like cancer cells, contain large concentrations of topoisomerases, which, therefore, serve as obvious therapeutic targets in cancer treatment. Drugs with topoisomerase II inhibition properties are widely used in chemotherapy, whereas topoisomerase I inhibitors have proven clinically useful in the treatment of colon cancer. Topoisomerase inhibition has not been reported for any of the naturally occurring phenazines but has been intensively pursued in synthetic analogues [97,100].

Radical scavenging is probably the most important ability associated with phenazines since it is believed that free radicals are involved in the development and progression of a wide range of serious human diseases. Oxygen-delivered free radicals are known to induce irreversible damage to neuronal cells in diseases like Parkinson’s disease and, perhaps, dementia, atherosclerosis, and cerebral traumas or strokes [101]. They are also believed to be involved in inducing carcinogenesis, aging of cells, asthma, renal failure, rheumatoid arthritis, and inflammation [101]. Natural antioxidants like vitamins C and E have multiple physiological functions but are insufficient in the treatment of free-radical-induced diseases. Therefore, efficient synthetic radical scavengers or antioxidants are needed to reduce these types of damage, including the irreversible loss of neural tissue. In normal cells, free radicals are continuously produced by metabolic enzymatic processes in the mitochondria as a part of energy production. Likewise, the body utilizes free radicals in immune responses toward incoming pathogens and in the degeneration of toxins. However, the overproduction of radicals or other dysfunctions causes the accumulation of radicals inside the cell and, consequently, cell death. The antioxidant or radical scavenging character of the phenazines can, therefore, be confirmed by measuring their ability to inhibit lipid peroxidation in liver microsomes [101]. Pyocyanin isolated from *P. aeruginosa* demonstrated higher scavenging activities at much lower concentrations than ascorbic acid [102].

Fungal and oomycete are common plant diseases, such as *Rhizoctonia solani Kuhn* and *Fusarium graminearum.* These fungus pathogens cause sheath blight of rice and blight of wheat generating huge losses during production and postharvest storage [103]. The intervention of synthetic agrochemicals is the most effective method at present; however, the current chemical-control agents are not fully effective at inhibiting these fungi’s activity. Further, it is necessary to develop more effective novel agents to replace conventional agrochemicals that introduce massive environmental pollutants and cause soil-borne diseases [104,105].

An effective method to replace agrochemicals is the development of new “green agrochemicals”, which are active compounds contained in natural products. As it was previously mentioned, several phenazines present activity against several fungi. The most common active phenazines are PCA, PCN, 1-hydroxyphenazine (1HP), 2-hydroxyphenazine (2HP), and 2-hydroxyphenazine-1-carboxylic acid (2HPCA), with the former being the most common [8]. The phenazine concentrations are detected on a scale of µg/mL, however, the concentration differs depending on the strain [106], and, usually, the concentration of PCA is present in a higher concentration.

Despite all known phenazines with antifungal activity, PCA is the only commercialized fungicide product with the trade name Shenqinmycin noted for its high fungicidal efficiency, low toxicity to humans and animals, friendliness to the environment, and improvement of crop production [103]. However, in the market, there are other products based on *Pseudomonas* strains, namely, *P. chlororaphis* which has been commercialized by companies from Europe (Cedomon BioAgri AB, Uppsala, Sweden) and the USA (AtEze, Turf Science Laboratories). Further, *P. fluorescens* A506 is the main component of BlightBAn A506 (Nufarm Americas Inc., Houston, TX, USA). These products have activity against fungi such as *Fusarium oxysporum*, *Aspergillus niger*, and *Botrytis cinerea* [8,33,85].

As described above, phenazines can be widely used as medicine and pesticides. However, due to the phenazine’s non-selective DNA binding, they may present toxic risks. Hence, Ou et al. [107] studied the degradation, adsorption, and leaching behaviors of a PCN in the agricultural soil of China. The authors found that PCN is easily degraded, has high adsorption affinity, and low mobility in high organic matter content and clay soils, thus, resulting in lower risks of contamination to groundwater systems. However, the risk of contamination increases greatly in soils containing low organic matter and low clay content leading to low adsorption affinity and moderate mobility in soils [107]. Studies like this should be performed to provide interesting insights concerning the use of phenazines.

## 8. Final Remarks

This review focused on the work developed in the field of natural phenazines, in particular, *Pseudomonas* spp. phenazines. In recent years, the research in this area has increased greatly, contributing to much clearer knowledge regarding the biosynthetic pathways and the regulatory mechanisms of phenazine production by bacteria. However, more research is needed to better understand the reason why one or another phenazine is produced.

Moreover, the natural phenazines showed exciting bioactivities, including antifungal properties against plant pathogens but also as an anticancer agent and as an antibiotic for human diseases caused by different pathogens. However, despite the interesting features of phenazines, only one PCA is commercialized as a biofungicide. Such limitation could be related to low phenazine concentrations produced by most bacterial strains. PCA is the only phenazine produced in higher concentrations that also shows the best biocontrol ability in comparison with the rest of the natural *Pseudomonas* spp. phenazines. Hence, more research is needed in order to increase phenazine production and in discovering new phenazines with higher production by bacteria and higher antifungal activity, since it is of major importance to substitute the synthetic agrochemical with natural pesticides, in order to reduce the environmental burden. Moreover, future studies should be focused on the phenazine production process scale-up, considering all the challenges that a process scale-up may have, namely, in the downstream process. Therefore, more studies should be performed to optimize extraction and purification methods in order to enable the process scale-up, but also to reduce the use of organic solvents (e.g., chloroform) and substitute them with greener solvents, for instance, ethyl acetate or DES (Deep Euthetic Solvents).

Further, despite this review being about pseudomonad phenazines, it is known that many other microorganisms are able to produce phenazines, thus, the research about phenazine production by such bacteria should increase.

## Figures and Tables

**Figure 1 molecules-28-01368-f001:**
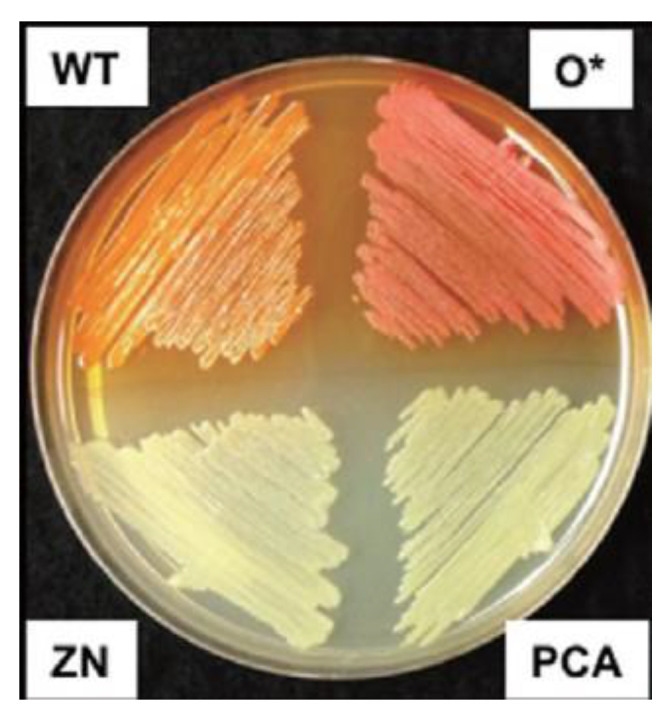
On agar plate, strains: *P. chlororaphis* 30-84 wildtype (WT), phenazine non-producer (ZN), PCA only producer (PCA), and 2-OH-PCA overproducer (O*) [18].

**Figure 2 molecules-28-01368-f002:**
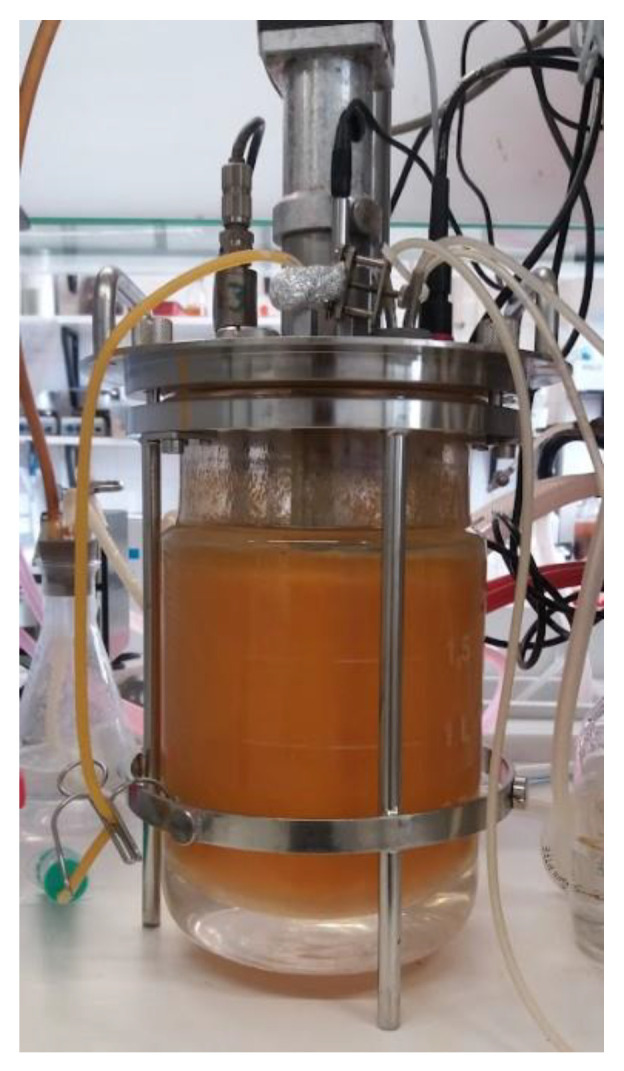
Phenazine production on a bioreactor cultivation of *Pseudomonas chlororaphis* subsp. *aurantiaca* DSM 19603.

**Figure 3 molecules-28-01368-f003:**
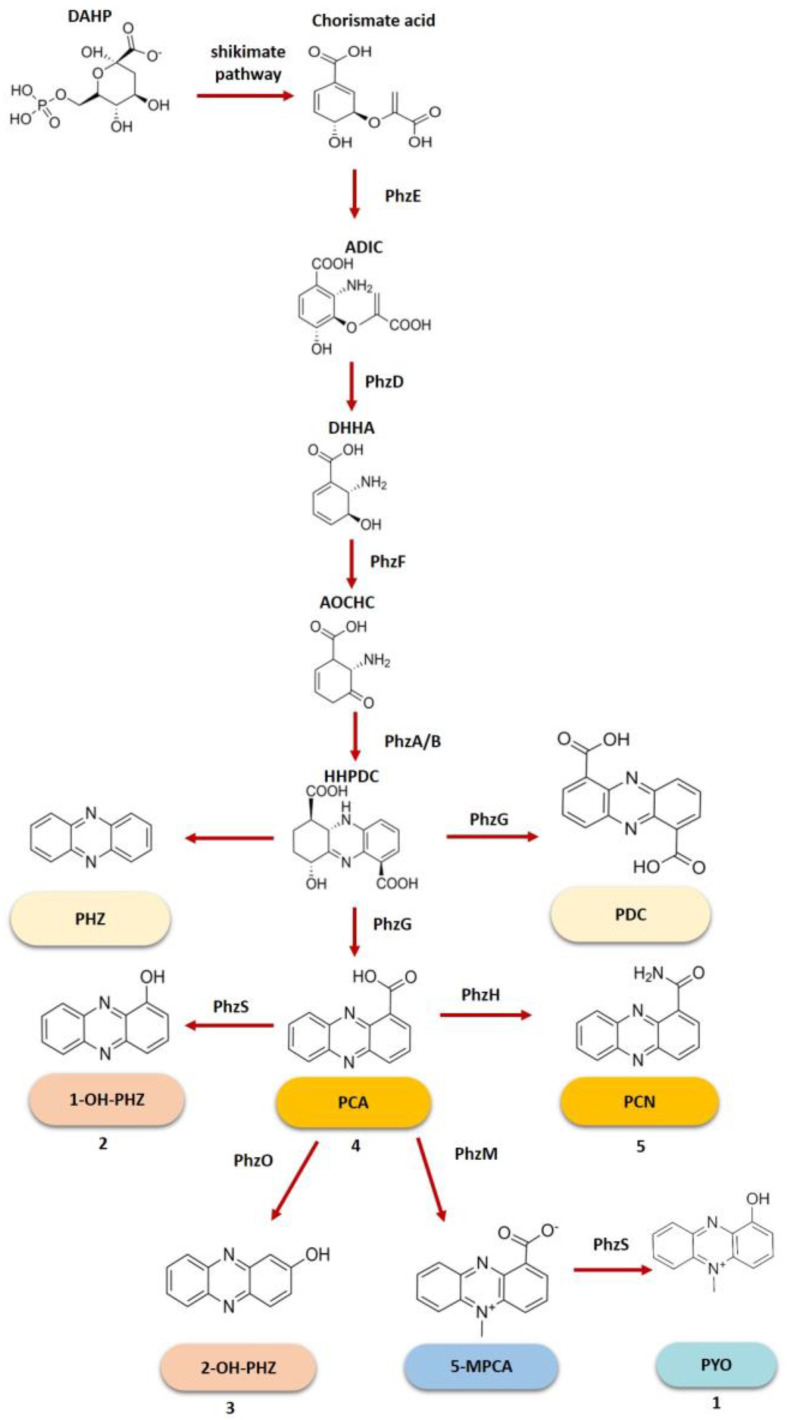
Biosynthetic pathway for phenazine production by *Pseudomonas* spp. DAHP: 3-deoxy-d-arabino-heptulosonate 7-phosphate; ADIC: 2-amino-2-deoxyisochorismate; DHHA: trans 2,3-dihydro-3-hydroxyanthranilic acid; AOCHC: 6-amino-5-oxocyclohex-2-ene-1-carboxylic acid; HHPDC: hexahydro-phenazine-1,6-dicarboxylate; PHZ: phenazines; PDC: phenazine-1,6-dicarboxylic acid; PCA: **4** phenazine-1-carboxylic acid; 1-OH-PHZ: **2** 1-hydroxy-phenazine; PCN: **5** phenazine-1-carboxamide; 2-OH-PHZ: **3** 2-hydroxy-phenazine; 5-MPCA: 5-methylphenazine 1-carboxylato; PYO: **1** pyocianin.

**Figure 4 molecules-28-01368-f004:**
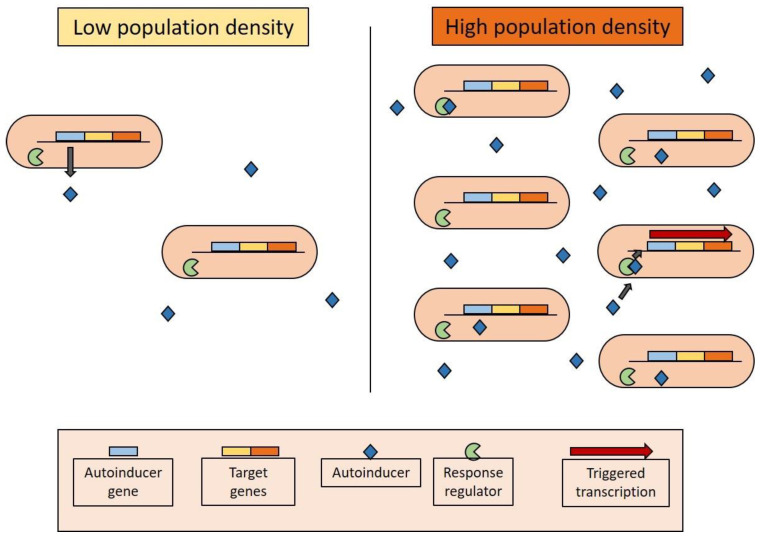
Quorum-sensing regulation mechanism.

**Figure 5 molecules-28-01368-f005:**
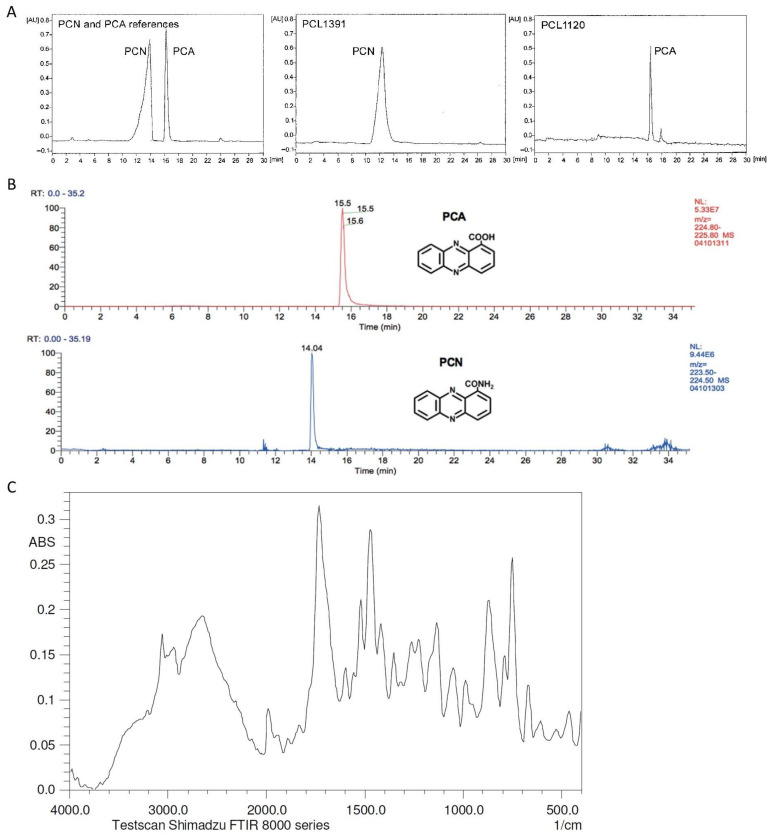
Spectra of different characterization analyses of phenazines, (**A**) Phase-reverse HPLC chromatogram of PCA and PCN [49]; (**B**) PCA and PCN peaks detected by LC-MS [18]; (**C**) FTIR spectrum of crystalline PCA [76].

**Figure 6 molecules-28-01368-f006:**
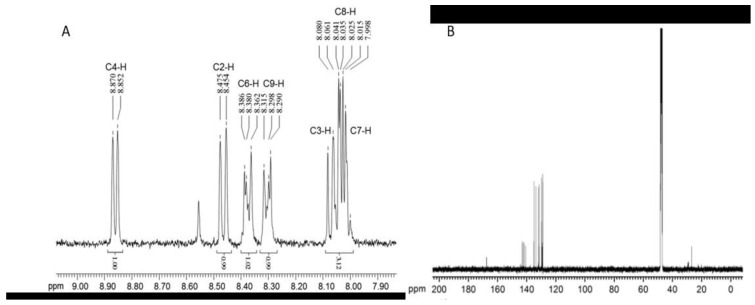
Spectra of PCN phenazine by (**A**) ^1^H NMR and by (**B**) ^13^C NMR [12].

## Data Availability

Not applicable.

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
