# Peer review of "Recent Developments in the Biological Activities, Bioproduction, and Applications of Pseudomonas spp. Phenazines"

_molecules, 2023, doi:10.3390/molecules28031368_

Round 1
Reviewer 1 Report
In my opinion, this review is too general, does not cover all the literature in the field and lacks of a critical point of view. The topics that have been covered are too diversed and are not well connected thouroguht the review. There is a significant lack of novelty since some reviews regarding the biocontrol ability of phenazines have been reported. Maybe, a review regarding phenazine production convering all the reported references would be more significant. In addition, there are numerous typing mistakes that make the review unreadable. For all these reasons, I can only recommend the rejection of the manuscript.
Author Response
Reviewer 1
- Extensive editing of English language and style required
Answer: The English language and style was edited throughout the manuscript.
- In my opinion, this review is too general, does not cover all the literature in the field and lacks of a critical point of view. The topics that have been covered are too diversed and are not well connected thouroguht the review. There is a significant lack of novelty since some reviews regarding the biocontrol ability of phenazines have been reported. Maybe, a review regarding phenazine production convering all the reported references would be more significant. In addition, there are numerous typing mistakes that make the review unreadable. For all these reasons, I can only recommend the rejection of the manuscript.
Answer: The authors acknowledge the Reviewer´s comments. However, in the manuscript a revision of the most important topics for Pseudomonas´ phenazines was performed. The comprised topics goes from the producing strains, biosynthetic pathways, followed by production process, extraction/ purification and characterization ending on the bioactivities and applications. Therefore, the authors think that the manuscript is well organized and not too general.
The novelty of this manuscript is focused on the literature review made for the phenazines production bioprocess subject, since as far as we know had never been revised in any previous review paper. Moreover, in this manuscript the characterization techniques reported in literature are also addressed, taking in consideration the papers published in the last years, another topic that authors did not find revised in any other publication.
The authors have also made some final remarks about phenazines low production yields and their bioactivities (lines 462 to 472). Nevertheless, we added more details to the discussion of the revised manuscript, which can be found between lines 472 and 478 “Moreover, future studies should be focused on the phenazines production process scale up considering all the challenges that a process scale up may have, namely in the downstream process. Therefore, more studies should be done to optimize the extraction and purification methods in order to enable the process scale up, but also to reduce the use of organic solvents (e.g. chloroform) and substitute them by greener solvents, for instance ethyl acetate or DES (Deep Euthetic Solvents).”

Reviewer 2 Report
The article is well designed and written. The study seems interesting, it could be accepted due to significance and further advantages compared to the existing literature.
I have only one substantive objection. I think that the work should mention the chapter 6 Phenazines Characterization should be extended, more spectra should be added for example in supplementary materials. This would be valuable for later publication citation
Author Response
Reviewer 2
The article is well designed and written. The study seems interesting, it could be accepted due to significance and further advantages compared to the existing literature.
- I have only one substantive objection. I think that the work should mention the chapter 6 Phenazines Characterization should be extended, more spectra should be added for example in supplementary materials. This would be valuable for later publication citation
Answer: The authors appreciate the comments of the Reviewer. Section six was extended and more spectra were added as supplementary material. Please see lines 343 and 362: “Jasim et al. [71] identified a PCA produced by an endophytic P. aeruginosa isolated from Zingiber officinale. Firstly, they extracted the compound and then was purified by TLC being obtained 7 different fraction, then the presence of compounds was measured by UV scan. Further bioactivity was tested and the fraction that showed activity against Pythyum myriotylum was subjected to LC-MS (Light Scattering – Mass Spectroscopy) to identify the compound (Figure S1).
Extracellular metabolites were extracted using ethyl acetate and a UV spectrum was recorded to quantify phenazines. Further the compound was purified using silica-gel column chromatography. The purified compound was characterized and identified using different techniques, namely FTIR, NMR and mass spectroscopy (supplementary material). IR spectra (Figure S2 A) showed peaks related the aromatic C–H stretching vibrations (2963, 2924 and 2853 cm−1). an amide carbonyl group was also confirmed. The 1H NMR spectrum (Figure S2 B) of the P. aeruginosa MML2212 purified compound showed two doublets, two protons were also displayed. Further, three aromatic protons appeared as a multiplet in the range. The 13C NMR (Figure S2 C) spectrum of the purified compound showed the presence of a carbonyl carbon of the amide group. Finally, the sample was subjected to mass spectroscopy (Figure S2 D); the compound was identified as PCN (phenazine-1-carboxamide) [74].
P. aurantiaca produced secondary metabolites with antifungal activity. The metabolites were identified using TLC, UV spectra, and MALDI-TOFF spectra and revealed to be PCA, 2-hydroxyphenazine (2-OH-PHZ), and N-hexanoyl homoserine lactone (HHL) (Figure S3) [73]”.
Supplementary spectra were added at the end of the paper document.

Reviewer 3 Report
This review of naturally occurring needs several improvements before it can be published.
Firstly, only structures of some of the piperazines have been shown. The compounds are denominated by strange abbreviations instead of numbers. The compound should be given with structures and numbers. The abbreviations could be added in parentheses. For rephrasing the chemical part of this review a chemist should be consulted.
I miss a definition of phenazine(s)
Secondly, all the methods mentioned for the preparation of different phenazines only can be performed in a small scale. If the compounds should be used for drugs or pesticides a large-scale procedure for production must be developed.
Thirdly, A number of biological activities are mentioned but nowhere in what concentration are these activities observed. If the activities are only seen in high concentration, they are of no interest. The concentration range should be compared to the range of positive reference compounds.
Fourthly a linguistic revision is needed as illustrated below for some but not all examples:
Pseudomonals’ phenazines, phenazines produced by pseudomonas.
Line 23 The first phenazines were prepared by..
Fig. 3. A pentavalent caron in the lowest structure. The figure is very difficult to read even after magnification.
Legend Fig. 6 1H NMR not H1 NMR. 13C NMR should be included.
Line 322-324. I do not understand this sentence.
333 why does the fungicidal effect exclude antibacterial?
Author Response
Reviewer 3
This review of naturally occurring needs several improvements before it can be published.
- Firstly, only structures of some of the piperazines have been shown. The compounds are denominated by strange abbreviations instead of numbers. The compound should be given with structures and numbers. The abbreviations could be added in parentheses. For rephrasing the chemical part of this review a chemist should be consulted.
Answer: Authors understand the comment of the Reviewer, although the denomination used by the authors was the one used in the different cited references focused on microbial phenazines. The chemical structures of the different phenazines are in the Figure 3.
- I miss a definition of phenazine(s)
Answer: The definition of phenazines are stated in lines 21 and 22: “Phenazines are a large group of heterocyclic, coloured and nitrogen-containing compounds of biological and chemical origin with great stability in natural environments”.
- Secondly, all the methods mentioned for the preparation of different phenazines only can be performed in a small scale. If the compounds should be used for drugs or pesticides a large-scale procedure for production must be developed.
Answer: Further details regarding process scale up were added to the Final remarks section, namely between lines 472 and 478: “Moreover, future studies should be focused on the phenazines production process scale up considering all the challenges that a process scale up may have, namely in the downstream process. Therefore, more studies should be done to optimize the extraction and purification methods in order to enable the process scale up, but also to reduce the use of organic solvents (e.g. chloroform) and substitute them by greener solvents, for instance ethyl acetate or DES (Deep Euthetic Solvents)”.
- Thirdly, A number of biological activities are mentioned but nowhere in what concentration are these activities observed. If the activities are only seen in high concentration, they are of no interest. The concentration range should be compared to the range of positive reference compounds.
Answer: The minimum inhibitory concentration at 50% and 90 % (MIC50 and MIC90) for some phenazines were added to the manuscript, between lines 350 and 353 “Moreover, future studies should be focused on the phenazines production process scale up considering all the challenges that a process scale up may have, namely in the downstream process. Therefore, more studies should be done to optimize the extraction and purification methods in order to enable the process scale up, but also to reduce the use of organic solvents (e.g. chloroform) and substitute them by greener solvents, for instance ethyl acetate or DES (Deep Euthetic Solvents)”. And between lines 357 and 361 “D-Alanylgriseoluteic acid, a potent antimicrobial phenazine compound is a prime example of the antibacterial activity of these metabolites. This phenazine exhibit a MIC50 value of 0.25 µg/mL and MIC90 value of 0.5 µg/mL for penicillin resistant isolates of Streptococcus pneumoniae. Both of MIC values of penicillin for these isolates were 4 µg/mL [92]”.
- Fourthly a linguistic revision is needed as illustrated below for some but not all examples:
Answer: The manuscript was detailed revised about the English language.
Pseudomonas’ phenazines, phenazines produced by pseudomonas.
Answer: Corrections were done to the manuscript regarding this comment.
Line 23 The first phenazines were prepared by..
Answer: This sentence is about the first time that a natural phenazine was observed in Nature. The sentence was changes talking into consideration the Reviewer comment: “These natural compound were firstly referred in 1859 by Fordos [1,2]….”.
Fig. 3. A pentavalent caron in the lowest structure. The figure is very difficult to read even after magnification.
Answer: The figure has been corrected.
Legend Fig. 6 1H NMR not H1 NMR. 13C NMR should be included.
Answer: The legend has been corrected.
Line 322-324. I do not understand this sentence.
Answer: The sentence has been corrected: “Kern and Newman [83] reported the use of HPLC to separate the different phenazines, namely PYO, PCA and 1-hydroxyphenazine (1-OH-PHZ) and then were quantified by UV absorption.” – Line 327 - 329.
Line 333 why does the fungicidal effect exclude antibacterial?
Answer: The fungicidal effect and antibacterial are different effect, but phenazines could have fungicidal and antibacterial activity, i.e. the same phenazine could present activity against some fungus and against some bacteria.
Round 2
Reviewer 3 Report
The authors have to some extent answered my questions. However, I do not find answers to the below questions.
The definition of phenazine given in lines 21 and 22 is incorrect. A definition could be a systematic name of the phenazine nucleus.
Biological activities I see no numbers of biological activities lines 350 to 361. Maybe the authors think of lines 380 and 387? In contrast, I see biological activities lines 393 to 402 but no numbers for the activities.
The authors maintain abbreviations instead of compound numbers in the schemes. I guess this now is an editorial decision if these are accepted.
Author Response
As recommended by the reviewer compound numbers have been added to the manuscript.